# ADAPTIVE POSTERIOR LEARNING:
# FEW-SHOT LEARNING WITH A SURPRISE-BASED MEMORY MODULE

**Tiago Ramalho**
Cogent Labs
tramalho@cogent.co.jp

**Marta Garnelo**
DeepMind
garnelo@google.com

## ABSTRACT

The ability to generalize quickly from few observations is crucial for intelligent systems. In this paper we introduce APL, an algorithm that approximates probability distributions by remembering the most surprising observations it has encountered. These past observations are recalled from an external memory module and processed by a decoder network that can combine information from different memory slots to generalize beyond direct recall. We show this algorithm can perform as well as state of the art baselines on few-shot classification benchmarks with a smaller memory footprint. In addition, its memory compression allows it to scale to thousands of unknown labels. Finally, we introduce a meta-learning reasoning task which is more challenging than direct classification. In this setting, APL is able to generalize with fewer than one example per class via deductive reasoning.

## 1 INTRODUCTION

Consider the following sequential decision problem: at every iteration of an episode we are provided with an image of a digit (e.g. MNIST) and an unknown symbol. Our goal is to output a digit $Y = X + S$ where $X$ is the value of the MNIST digit, and $S$ is a numerical value that is randomly assigned to the unknown symbol at the beginning of each episode. After seeing only a single instance of a symbol an intelligent system should not only be able to infer the value $S$ of the symbol but also to correctly generalize the operation associated with the symbol to any other digit in the remaining iterations of that episode.

Despite its simplicity, this task emphasizes three cognitive abilities that a generic learning algorithm should display: 1. the algorithm can learn a behaviour and then flexibly apply it to a range of different tasks using only a few context observations at test time; 2. the algorithm can memorize and quickly recall previous experiences for quick adaptation; and 3. the algorithm can process these recalled memories in a non-trivial manner to carry out tasks that require reasoning.

The first point is commonly described as "learning to learn" or *meta-learning*, and represents a new way of looking at statistical inference (Schmidhuber, 1987; Bengio et al., 1990; Bengio & LeCun, 2007). Traditional neural networks are trained to approximate arbitrary probability distributions with great accuracy by parametric adaptation via gradient descent (LeCun et al., 2015; Schmidhuber, 2015). After training that probability distribution is fixed and neural networks can only generalize well when the testing distribution matches the training distribution (Neyshabur et al., 2017). In contrast, meta-learning systems are trained to learn an *algorithm* that infers a function directly from the observations it receives at test time. This setup is more flexible than the traditional approach and generalizes better to unseen distributions as it incorporates new information even after the training phase is over. It also allows these models to improve their accuracy as they observe more data, unlike models which learn a fixed distribution.

The second requirement - being able to memorize and efficiently recall previous experience - is another active area of research. Storing information in a model proves especially challenging as we move beyond small toy-examples to tasks with higher dimensional data or real-world problems.

Current methods often work around this by summarizing past experiences in one lower-dimensional representation (Hochreiter & Schmidhuber, 1997; Kingma & Welling, 2013) or using memory modules (Graves et al., 2016). While the former approach can produce good results, the representation and therefore the amount of information we can ultimately encode with such models will be of a fixed and thus limited size. Working with neural memory modules, on the other hand, presents its own challenges as learning to store and keep the right experiences is not trivial. In order to successfully carry out the task defined at the beginning of this paper a model should learn to capture information about a flexible and unbounded number of symbols observed in an episode without storing redundant information.

Finally, reasoning requires processing recalled experiences in order to apply the information they contain to the current data point being processed. In simple cases such as classification, it is enough to simply recall memories of similar data points and directly infer the current class by combining them using a weighted average or a simple kernel (Vinyals et al., 2016; Snell et al., 2017), which limits the models to performing interpolation. In the example mentioned above, more complex reasoning is necessary for human-level generalisation.

In this paper we introduce Approximate Posterior Learning (APL, pronounced like the fruit), a self-contained model and training procedure that address these challenges. APL learns to carry out few-shot approximation of new probability distributions and to store only as few context points as possible in order to carry out the current task. In addition it learns how to process recalled experiences to carry out tasks of varying degrees of complexity. This sequential algorithm was inspired by Bayesian posterior updating (Jaynes, 2003) in the sense that the output probability distribution is updated as more data is observed.

We demonstrate that APL can deliver accuracy comparable to other state-of-the-art algorithms in standard few-shot classification benchmarks while being more data efficient. We also show it can scale to a significantly larger number of classes while retaining good performance. Finally, we apply APL to the reasoning task introduced as motivation and verify that it can perform the strong generalization we desire.

The main contributions of this paper are:

- A simple memory controller design which uses a surprise-based signal to write the most predictive items to memory. By not needing to learn what to write, we avoid costly backpropagation through memory which makes the setup easier and faster to train. This design also minimizes how much data is stored, making our method more memory efficient.

- An integrated external and working memory architecture which can take advantage of the best of both worlds: scalability and sparse access provided by the working memory; and all-to-all attention and reasoning provided by a relational reasoning module.

- A training setup which steers the system towards learning an algorithm which approximates the posterior without backpropagating through the whole sequence of data in an episode.

## 2 MODEL

### 2.1 ARCHITECTURE

Our proposed model is composed of a number of parts: an **encoder** that generates a representation for the incoming query data; an external **memory store** which contains previously seen representation/ data pairings with writing managed by a **memory controller**; and a **decoder** that ingests the query representation as well as data from the memory store to generate a probability distribution over targets. We describe each of the parts in detail below[1].

**Encoder** The encoder is a function which takes in arbitrary data $x_t$ and converts it to a representation $e_t$ of (usually) lower dimensionality. In all our experiments $x_t$ is an image, and we therefore choose a convolutional network architecture for the encoder. Architectural details of the encoder used for each of the experiments are provided in the appendix.

---

[1]Source code for the model is available at `https://github.com/cogentlabs/apl`.

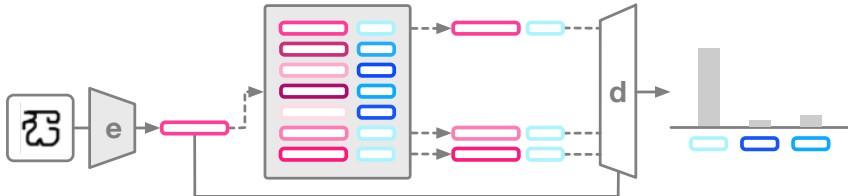

Figure 1: APL model applied to the the classification of an Omniglot image. The encoded image is compared to the entries of the memory and the most relevant ones are passed through a decoder that outputs a probability distribution over the labels. The dotted line indicates the parts of the graph that are not updated via back-propagation.

**Memory store** The external memory module is a database containing the stored experiences. Each of the columns corresponds to one of the attributes of the data. In the case of classification, for example, we would store two columns: the embedding $e_m$ and the true label $y_m$. Each of the rows contains the information for one data point. The memory module is queried by finding the k-nearest neighbors between a query and the data in a given column. The full row data for each of the neighbors is returned for later use. The distance metric used to calculate proximity between the points is an open choice, and here we always use euclidean distance.

Since we are not backpropagating through the memory, how do we ensure that the neighbors returned by the querying mechanism contain task-relevant information? We expect that class-discriminative embeddings produced by the encoder should cluster together in representation space, and therefore should be close in the sense of euclidean distance. While this is not mathematically necessary, in the following sections we will show that APL as proposed does work and retrieve the correct neighbors which means that in practice our intuition holds true.

**Memory controller** We use a simple memory controller which tries to minimize the amount of data points written to memory. Let us define surprise as the quantity associated with the prediction for label $y_t$ as $\mathbf{S} = -ln(y_t)$. Intuitively, this means that the higher the probability our model assigns to the true class, the less surprised it will be.

This suggests a way of storing the minimal amount of data points in memory which supports maximal classification accuracy. If a data point is 'surprising', it should be stored in memory; otherwise it can be safely discarded as the model can already classify it correctly.

How should the memory controller decide whether a point is 'surprising'? In this work we choose the simplest possible controller: if the surprise is greater than some hyperparameter $\sigma$, then that data should be stored in memory. For our experiments, we choose $\sigma \propto -\ln(N)$ where $N$ is the number of classes under classification which means that if the prediction confidence in the correct class is smaller than the probability assigned by a uniform prediction the value should be written to memory. In the appendix we show that after model training model performance is robust to variations in $\sigma$, as surprise becomes highly bimodal: a new data point tends to be either highly surprising (never seen something similar before) or not very surprising.

Conveniently, in the case of classification problems the commonly used cross-entropy loss reduces to our measure of surprise directly, and we therefore use the prediction loss as an input to the memory controller directly.

**Decoder** The decoder takes as input the query representation as well as all the data from the neighbors found in the external memory. We designed a relational feed-forward module with self attention which takes particular advantage of the external memory architecture. In addition we tested two other established decoder architectures: an unrolled relational working memory core and an unrolled LSTM. As all experiments have a classification loss at the end, all the decoders return a vector with logits for the $N$ classes under consideration. Full details of each architecture are provided in the Appendix.

- **Relational self-attention feed-forward decoder.** The relational feed-forward module (see figure 2, left) processes each of the neighbors individually by comparing them with the

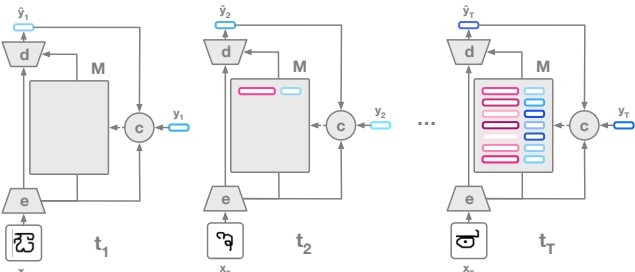

Figure 2: Two of the three different decoder architectures of APL: relational feed-forward memory (left) and relational working memory (Santoro et al., 2018) (right). In the figure $e_t$ corresponds to the encoded target, $e_{1...m}$ to the encoded observations from the memory, $l_{1...m} = f(y_{1...m})$ are the labels processed by an embedding layer and $d_{1...m}$ is the distance between $e_t$ and $e_{1...m}$.

query, and then does a cross-element comparison with a self-attention module before reducing the activations with an attention vector calculated from neighbor distances.

- **Relational working memory decoder** (Santoro et al., 2018) The relational working memory module (figure 2, right) takes in the concatenated neighbor embeddings and corresponding label embeddings as its initial memory state. The query is fed a number $N$ times as input to the relational memory core to unroll the computation.

- **LSTM decoder** Finally we also test a vanilla LSTM decoder that takes in the query as the initial memory state and is fed each of the concatenated neighbor embeddings and corresponding label embeddings as its input each time step.

## 2.2 TRAINING SETUP

Since we are looking for a system which can update its beliefs in an online manner we need a training procedure that reflect this behaviour. We train the system over a sequence of *episodes* that are composed of sequences of pairs $(x_t, y_t)$. At the start of every episode the mapping $x_t \rightarrow y_t$ is shuffled in a deterministic manner (the exact details are task dependent and will be outlined in the experiments section). The data is then presented to the model sequentially in a random order. The model's memory is empty at the beginning of the episode.

At each time step, a batch of examples is shown to the model and a prediction is made. We then measure the instantaneous loss $L(\hat{y}_t, y_t)$ and perform a gradient update step on the network to minimize the loss on that batch alone. The loss is also fed to the memory controller for the network to decide whether to write to memory. In all the experiments below the task is to classify some quantity, therefore we use cross entropy loss throughout.

Figure 3: APL training over several iterations. The encoder $e$ embeds the query image that is compared against the stored experiences in the memory $M$. Matches are fed alongside the encoded image into a decoder $d$. Finally, the controller $c$ decides whether the currently observed example should be stored in memory.

APL learns a sequential update algorithm, that is, it minimizes the expected loss over an episode consisting of a number of data elements presented sequentially. However we don't need to back-propagate through the sequence to learn the algorithm. Rather, the model's parameters are updated to minimize the cross-entropy loss independently at each time step. Therefore the only pressure to

learn a sequential algorithm comes from the fact that episodes are kept small so that the decoder is encouraged to read the information coming from the queried neighbors instead of just learning to fit the current episode's label mapping in its weights after a few steps of gradient descent (which is what happens in the case of MAML (Finn et al., 2017)).

## 3 RELATED WORK

Meta-learning as a research field covers a large number of areas. The concept of 'learning to learn' is not tied to a specific task and thus meta-learning algorithms have been successfully applied to a wide range of challenges like RL (Wang et al., 2016; Finn et al., 2017), program induction (Devlin et al., 2017) few-shot classification (Koch et al., 2015; Vinyals et al., 2016) and scene understanding (Eslami et al., 2018).

Some meta learning models generate predictions in an autoregressive fashion by predicting the next target point from the entire prior sequence of consecutive observations (Reed et al., 2018; Mishra et al., 2018). Algorithms of this kind have delivered state-of-the art results in a range of tasks such as supervised learning to classification. Nonetheless their reliance on the full context history in addition to their autoregressive nature hinders parallelization and hurts performance and scalability.

Another set of methods is based on the nearest neighbours approach (Koch et al., 2015; Vinyals et al., 2016; Snell et al., 2017). These methods use an encoder to find a suitable embedding and then perform a memory look up based on these representations. The result is a weighted average of the returned labels. As shown in (Mishra et al., 2018) using a pure distance metric to compare neighbors results in worse performance than allowing a network to learn a comparison function. These kinds of methods thus suffer in comparison. Meta Networks (Munkhdalai & Yu, 2017) also use an external memory to enable learning from previous examples combined with a model featuring slow and fast weights to produce the output, enabling them to state-of-the-art performance in several benchmarks.

Conditional neural processes (Garnelo et al., 2018) summarize the data into a fixed representation by averaging over the outputs of an encoder. This representation is fed into a decoder together with a query to produce the output. These methods are more space and compute efficient but given the fixed and averaged representation may not scale to very large problems.

All of the above methods expect a fixed size context, thereby making life-long learning over large time horizons difficult. To enable this, an algorithm must learn to only write observations into memory when they provide additional predictive power. Memory augmented neural networks (MANN) (Santoro et al., 2016) achieve this by learning a controller to write into a differentiable neural dictionary. However, this requires backpropagating through the entire sequence to learn, which makes credit assignment over long time sequences hard and is computationally expensive. The idea of using an external memory module has been explored in (Kaiser et al., 2017) and shown to produce good results. Compared to that work we introduce a simpler writing mechanism and the idea of a relational decoder to exploit the nearest neighbor structure.

## 4 EXPERIMENTS

### 4.1 FEW-SHOT OMNIGLOT CLASSIFICATION

The Omniglot dataset contains 1623 characters with 20 examples each. 1200 of the character classes are assigned to the train set while the remaining 423 are part of the test set. The examples are presented to the model sequentially in batches of 16 examples. For each episode, we choose $N$ classes and shuffle their labels. We then run an episode for a certain number of steps, which is decreased as the model's accuracy increases to encourage quick adaptation. This means that the model accuracy and number of memories written to memory is time dependent.

We follow an architecture similar to those used in previous work for the image encoder (Vinyals et al., 2016; Mishra et al., 2018), which consists of four convolutional blocks with 3x3 convolutions, relu and batch normalization. We augment the number of classes in the training set by rotating each symbol 90, 180 and 270 degrees as in previous work (Santoro et al., 2016). For this task, we found that all three decoder architectures perform similarly. A detailed comparison and all hyperparameters needed to reproduce this experiment are provided in the Appendix.

In Figure 4a we can see the behavior of the algorithm within a single episode. As it sees more examples its performance increases until it saturates at some point, when additional writes don't help anymore (assuming the exact same data piece won't be seen again, which is the regime we always assume here). In the simple case of 5-way Omniglot classification, fewer than 2 examples per class are sufficient to saturate performance.

In Figure 4b we demonstrate the evolution of the posterior distribution in 20-way classification for 3 different, fixed inputs. For the first step, where the memory is empty APL learns to output a uniform distribution ($p \simeq 1/N$ with N the number of classes under classification). As more examples are added to memory, its distribution refines until it sees an informative example for that class, at which point its prediction becomes very confident.

In Figure 4c we can see that different numbers of examples are written to memory for different classes, which demonstrates one of the advantages of this framework: **we only need to store as many examples as each class requires**, therefore if some classes may be more easily classified than others we can optimally use memory. In contrast, other models feed in a fixed context size per class which means they will either suffer in accuracy or use excessive memory.

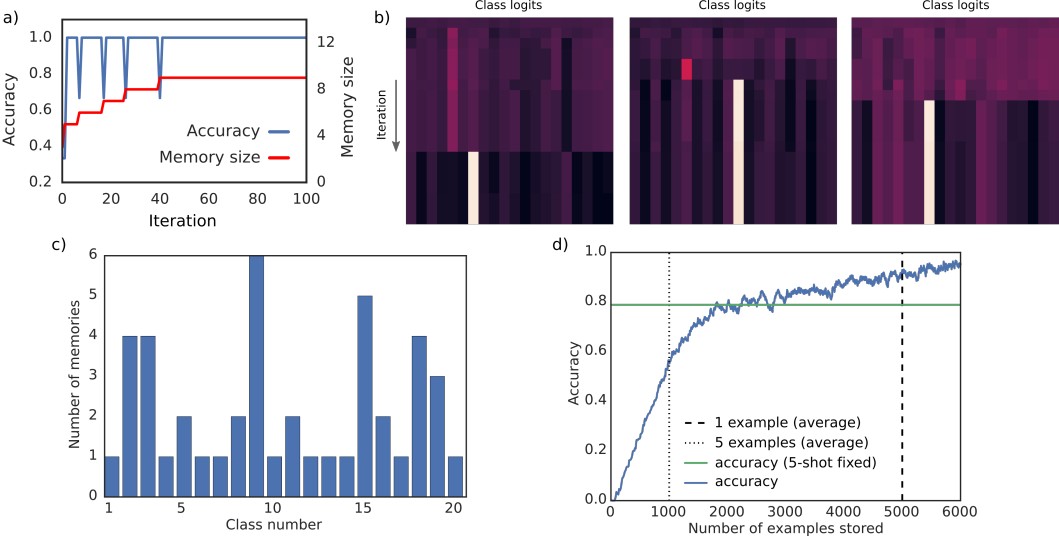

Figure 4: a) Accuracy and size of memory for 5-way Omniglot. APL stops writing to memory after having 2 examples per class for 5-way classification. b), examples of evolution of posterior distribution for 20-way classification for 3 images. The distribution starts as uniform for the very first step, then starts to change as more items are added to memory. When the correct class is seen, the distribution converges. c), the number of labels stored in memory per class is highly heterogeneous. In this 20-way problem, APL stored 44 items in memory and achieved 98.5% accuracy, which is higher than its homogeneous 5-shot accuracy. d) Accuracy vs. number of items written to memory for 1000-way classification. Classification accuracy when only 2000 examples have been written to memory (on average 2 examples per class) surpasses the accuracy for a fixed context size of 5 examples per class.

Despite the previous point, it is also worthwhile comparing model accuracy to existing baselines with a fixed context size. To this end, we pre-populate the memory of a trained model with 1 or 5 examples per class and calculate the accuracy over the test set. We emphasize that the model was not trained to do well in this fixed-context scenario, and yet for 1 and 5-shot classification we obtain performance comparable to state-of-the-art models without having extensively tuned hyperparameters. Furthermore we tested our model with a much higher number of classes: 423-way classification where where we can use the whole test set, and 1000-way where the test set is augmented with rotations of the characters as we do in training.

Finally, we test how the model fares on a completely new distribution, MNIST. For 1-shot, 10-way MNIST, APL trained on 20-way omniglot classification obtains 61% accuracy (compared to 72% cited by (Vinyals et al., 2016)). Testing our model in the sequential regime, we observe that it

| | 5-way | | 20-way | | 423-way | | 1000-way[†] | |
|---|---|---|---|---|---|---|---|---|
| | 1-shot | 5-shot | 1-shot | 5-shot | 1-shot | 5-shot | 1-shot | 5-shot |
| Matching nets | 98.1% | 98.9% | 93.8% | 98.5% | - | - | - | - |
| CNP | 95.3% | 98.5% | 89.9% | 96.8% | - | - | - | - |
| MANN | 82.2% | 94.9% | - | - | - | - | - | - |
| MAML | 98.7% | 99.9% | 95.8% | 98.9% | - | - | - | - |
| SNAIL | 99.07% | 99.78% | 97.64% | 99.35% | - | - | - | - |
| **APL** | 97.9% | 99.9% | 97.2% | 97.6% | 73.5% | 88.0% | 68.9% | 78.9% |

Table 1: Omniglot test accuracies for fixed context sizes compared to other baselines. ([†] For 1000-way classification rotated pseudoclasses are used.)

continues writing examples to memory as it sees surprising observations, which allows it to correct for the distribution shift. After writing 45 examples per class, it reaches an accuracy of 86% (there are 1000 examples per class in the MNIST test set, so it is not simply memorizing).

## 4.2 IMAGENET

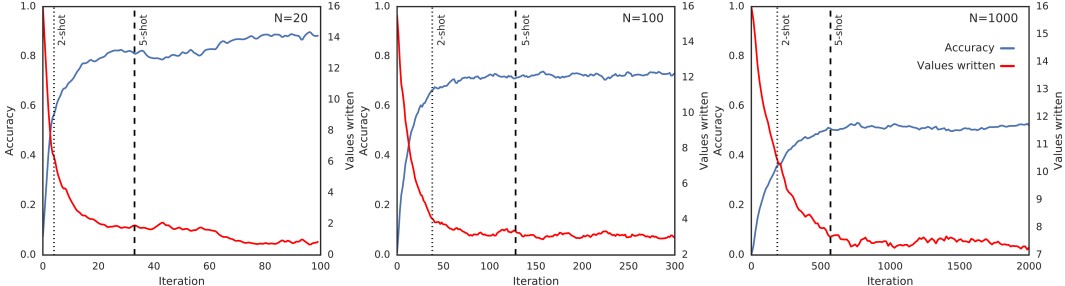

Figure 5: Evolution of top-1 accuracy and number of written examples to memory over a single episode for the Imagenet dataset. Curves are averages over 5 test episodes and smoothed with exponential moving average.

We also applied our model to the full scale imagenet dataset. Unlike the above experiments there are no held out classes for testing, as the dataset was not conceived with held out classes in mind. Instead, we rely on shuffling the labels amongst the 1000 Imagenet classes and using the images from the test set for evaluation. This means the generalization results are slightly weaker than in the above sections, but they still provide important insights as to the scalability of our method to thousands of classes and applicability to harder scenarios.

As an encoder we use the pretrained Inception-ResNet-v2 (Szegedy et al., 2017) due to computational constraints. For the fixed label case, this network reaches a top-1 accuracy of $80.4\%$. Training the encoder end-to-end might produce better results, an investigation which we leave to later work.

In the 20-way classification challenge, our method reaches $86.7\%$ top-1 accuracy (average accuracy after 50 iterations). Performance remains very high for 100-way ($72.9\%$ top-1 accuracy). The model's performance degrades somewhat ($52.6\%$ top-1 accuracy) for 1000-way classification, where all the classes are shuffled. This highlights that large scale meta-learning on real world datasets remains a challenge even when all the classes have been observed by the encoder, as in this case.

## 4.3 NUMBER ANALOGY

The number analogy task challenges a meta-learning model to use logic reasoning to generalize with fewer than 1 example per possible class (Figure 6). At each time step the network is shown two pieces of data, a number $X$ and a symbol $S$. It is asked to classify the result of $X + S$ based only on the current data and its previous experiments. We experiment with two levels of difficulty for this task: in the first, the number values are fixed and correspond to the MNIST digits, while there are

10 different symbols with unknown values in each episode; in the second, both digits and symbols have shuffled values. We sample the symbol values in the range $[-10, 10]$.

When querying the memory, we query $k$ neighbors via the number embeddings and $k$ neighbors via the symbol embeddings. This makes sure that any relevant information to the problem is available to the reasoning module. The rest of the training setup is identical to the Omniglot experiments, including the encoder network for the digits.

In the case where the numbers are fixed a human would need only see 10 examples, one for each symbol, to be able to correctly generalize for all 100 possible combinations. With one example per symbol, APL reaches 97.6% accuracy on the test set.

When both numbers and symbols are shuffled each episode, a logical deduction process must be performed to infer the correct symbols. Our model is able to generalize using 50 examples written to memory (Figure 6) which is still fewer than seeing one of all 100 possible combinations.

In this complex case, once a symbol's meaning has been figured out it is no longer necessary to solve the system of equations for that unknown. It would be interesting to explore how a system could additionally store this information in memory for later reuse, a question we leave for later work.

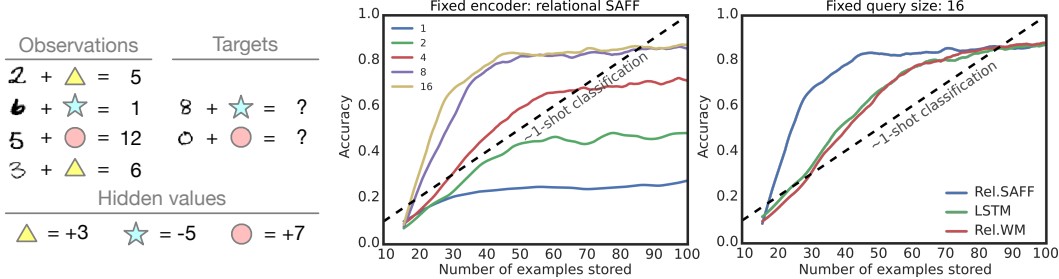

Figure 6: Left: Number analogy task. The colored symbols have unknown values that are consistent throughout an episode. Right: Accuracy as a function of number of examples seen for the Analogy task where the number meanings are also shuffled each episode. Curves are averages over 10 test episodes and smoothed with exponential moving average. On the left we fix the decoder (relational self-attention feed-forward module) and vary $k$. As there are 100 possible combinations of symbols (10 numbers $\times$ 10 symbols), the thick dashed line corresponds to the performance of a model capable of perfect 1-shot generalization. We can see that for $k = 8$ and $k = 16$ the decoder can infer the symbol and number meanings to do better than direct 1-shot classifications. On the right we fix $k = 16$ and show that the relational self-attention feed-forward module can generalize better from few examples than other decoder architectures.

## 5 CONCLUSION

We introduced a self-contained system which can learn to approximate a probability distribution with as little data and as quickly as it can. This is achieved by putting together the training setup which encourages adaptation; an external memory which allows the system to recall past events; a writing system to adapt the memory to uncertain situations; and a working memory architecture which can efficiently compare items retrieved from memory to produce new predictions.

We showed that the model can:

- Reach state of the art accuracy with a smaller memory footprint than other meta-learning models by efficiently choosing which data points to remember.
- Scale to very large problem sizes thanks to the use of an external memory module with sparse access.
- Perform fewer than 1-shot generalization thanks to relational reasoning across neighbors.

ACKNOWLEDGMENTS

The authors thank Elco Bakker, Alex Pritzel and David Raposo for insightful discussions; Paul Komarek, Adrià Puigdomènech for help with writing code; and Kevin McKee for revising the manuscript.

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

## 6 SUPPLEMENTARY MATERIAL

### 6.1 MODEL ARCHITECTURES AND TRAINING DETAILS

For all experiments below we use the same training setup. For each training episode we sample elements from $N$ classes and randomly shuffle them to create training batches. For every batch shown to the model, we do one step of gradient descent with the Adam optimizer. We anneal the learning rate from $10^{-4}$ to $10^{-5}$ with exponential decay over 1000 steps (decay rate 0.9).

For all experiments, the query data is passed through an embedding network (described below in), and this vector is used to query an external memory module. The memory module contains multiple columns of data, one of which will be the key and the others will contain data associated with that key. The necessary columns for each experiment are outlined below. The memory size is chosen so that elements will never be overwritten in a single episode (i.e. memory size > batch size $\times$ number of iterations).

The returned memory contents as well as the query are fed to one of the decoder architectures described in section 6.2.

#### 6.1.1 OMNIGLOT

For omniglot we encode the images with a convolutional network composed of a single first convolution to map the image to 64 feature channels, followed by 12 convolutional blocks. Each block is made up of a step of Batch Normalization, followed by a ReLU activation and a convolutional layer with kernel size 3. Every three blocks the convolution contains a stride 2 to downsample the image. All layers have 64 features. Finally we flatten the activations to a 1D vector and pass it through a Layer Normalization function.

For all decoders we use a hidden dimensionality of 512, take their final state and pass it through a linear layer to generate the final logits for classification, using a Cross Entropy loss.

#### 6.1.2 IMAGENET

The encoder is a pretrained Inception-ResNet-v2 network (Szegedy et al., 2017) with the standard preprocessing as described in the paper. We use as embedding the pre-logit activations. All decoders use a hidden dimensionality of 1024. After the decoder step we take their final state and pass it through a linear layer to generate the final logits for classification, using a Cross Entropy loss.

#### 6.1.3 ANALOGY TASK

The encoder for MNIST uses the same convolutional network as described in the Omniglot section. The symbols are one-hot vectors for the first set of experiments. The memory is queried for neighbors both of the digit embeddings as well as the symbols, and found neighbors are concatenated and fed to the decoder.

The decoder process is identical to Omniglot. However, the classification target is now the one-shot encoded version of the result of the computation $X + S$, where $X$ is the digit value and $S$ is the symbol value. As $S \in [-5, 5]$, we sum 5 to all values to obtain valid one-hot encodings (which means there are 20 possible values in all).

### 6.2 DECODER ARCHITECTURES COMPARISON

#### 6.2.1 RELATIONAL SELF-ATTENTION FEED FORWARD MODULE

Consider the set $\{e_t, e_{1...m}, l_{1...m}, d_{1...m}\}$, where $e_t$ is the encoded target, $e_{1...m}$ the encoded observations from the memory, $l_{1...m} = f(y_{1...m})$ are the labels processed by a simple embedding layer to project the classes into a higher dimensional space, and $d_{1...m}$ are the euclidean distances between $e_t$ and $e_{1...m}$. By concatenating all these vectors, we have a set of inputs to a relational block (Battaglia et al., 2018).

This tensor (of shape [batch size, k, sum of all embeddings feature sizes]) is fed to what we call a relational self-attentional block: first the tensor is passed through a multihead attention layer (Figure

7), which compares all elements to each other and returns a new tensor of the same shape; then a shared nonlinear layer (ReLU, linear, layer norm) processes each element individually. The self-attentional blocks are repeated 8 times in a residual manner (the dimensionality of the tensor never changes).

Finally, we pass the distances between neighbors and query through a softmax layer to generate an attention vector which is multiplied with the activations tensor over the first axis (this has the effect of weighting closer memories more). The tensor is then summed over that first axis to obtain the final representation.

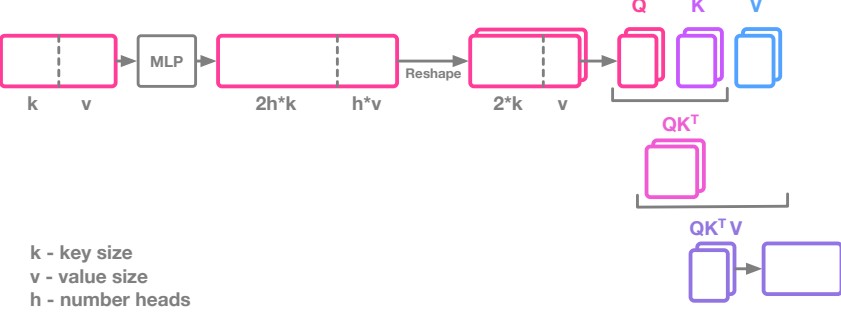

Figure 7: Multihead attention implementation.

### 6.2.2 RELATIONAL WORKING MEMORY

We use a Relational working memory core as described in (Santoro et al., 2018), (figure 2, right). The memory is initialized with the concatenated vectors $\{e_{1...m}, l_{1...m}, d_{1...m}\}$. The query $e_t$ is fed a number $N = 5$ times as input to the relational memory core to unroll the computation. The final memory state is passed through a linear layer to obtain the logits.

### 6.2.3 LSTM

We use a standard LSTM module, with initial state equal to the query embedding, and at each time step we feed in the neighbor embedding as concatenated with the embedded label. The LSTM is rolled out for $k$ time steps (i.e. the number of neighbors). Its final output state is taken as input to the logits linear layer.

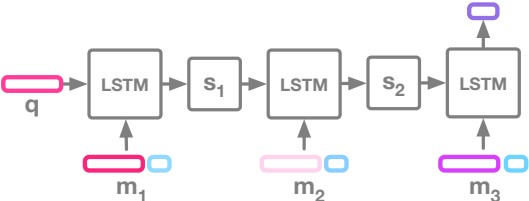

Figure 8: The LSTM decoder for APL.

### 6.2.4 DECODER ARCHITECTURE COMPARISON

We compared all three decoder architectures for the classification case and found they perform equally well for the classification case, as shown in the figure below.

For the analogy task, we found the Relational self-attention feed forward module to work best, as outlined in the main text.

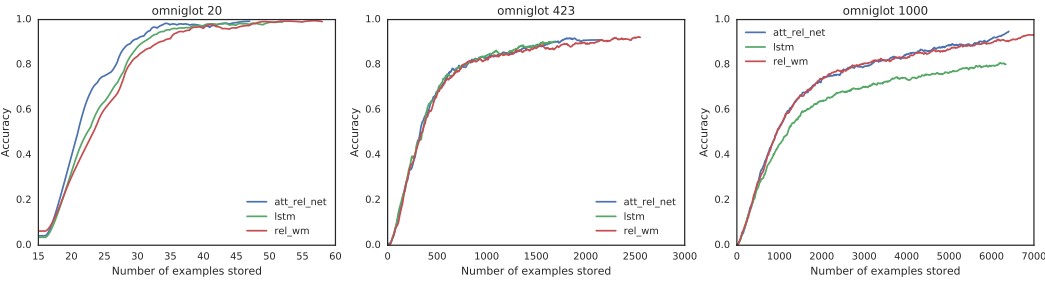

Figure 9: Accuracy as a function of examples written to memory. We compared relational working memory, LSTM and the relational self-attention feed-forward module for omniglot on the 20-way, 423-way and 1000-way tasks.

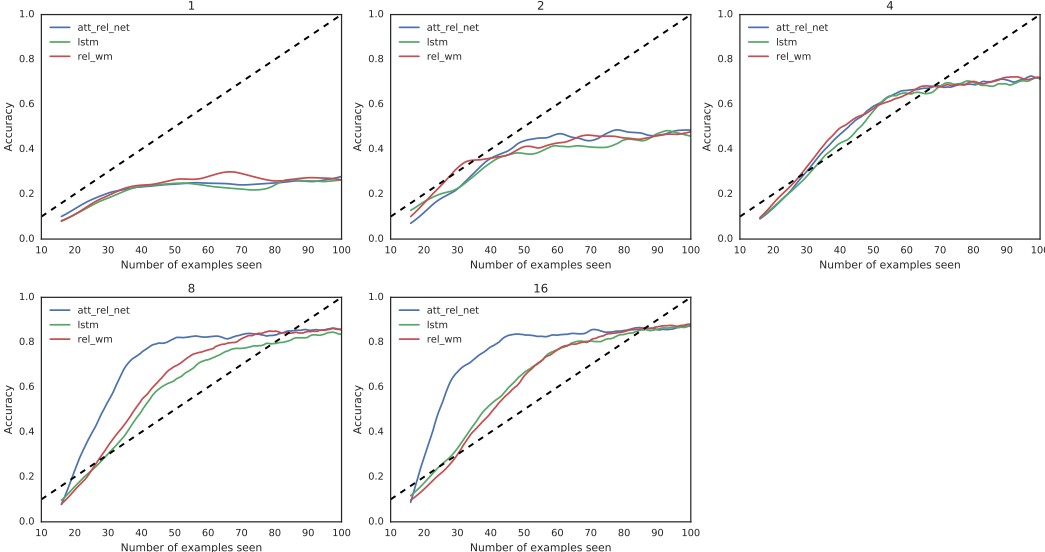

Figure 10: Accuracy as a function of examples written to memory. Each plot corresponds to a different number $k$ of retrieved nearest neighbors. We compared relational working memory, LSTM and the relational self-attention feed-forward module.

## 6.3 EFFECT OF THRESHOLD PARAMETER ON PERFORMANCE

How does the choice of parameter $\sigma$ affect the performance of APL? Empirically we have verified that for a large range of $\sigma$, memory size and accuracy are largely unchanged. This is due to the feedback loop between the number of items stored and classification accuracy: as more items are stored in memory, the more elements are correctly classified and not stored in memory. Therefore the memory storage mechanism is self-regulating, and the number of elements in memory ends up being largely flat.

In Fig. 11 we show the final memory size and average accuracy for the last 100 data points after showing APL 2000 unique data points for the case of 200-way classification. In this case the 'natural' (uniform predictions) $\sigma$ is around 5.2, which seems to be close to optimal for accuracy vs. elements in memory. We can increase the value somewhat but eventually the model can't write to memory any more and performance tanks. On the other side of the curve, for $\sigma = 0$ where we write everything, performance is slightly higher but at a roughly 8x memory cost.

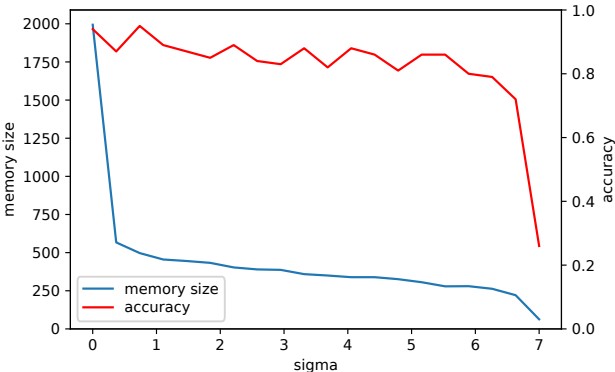

Figure 11: Accuracy as a function of examples written to memory. Each plot corresponds to a different number $k$ of retrieved nearest neighbors. We compared relational working memory, LSTM and the relational self-attention feed-forward module.

## 6.4 RELATION TO CONTINUAL LEARNING

While we study APL in the few-shot learning setting, the algorithm could also be used in the continual learning setup (Kirkpatrick et al., 2017; Rusu et al., 2016; Yoon et al., 2018; Rebuffi et al., 2017). We consider an experiment where each task consists of learning 10 new and previously unseen classes. For each task we present the models with 200 unique examples, and report the average accuracy for the last 100 examples seen. Examples are drawn from the test set of classes.

In the case of progressive networks, one gradient descent step is taken after each example. For each task, a new logits layer is added on top of a convolutional encoder (same architecuture as APL) pretrained on the omniglot training set. APL is run as described in the main text.

The results are summarized in figure 12: APL can perform as well or better than a progressive network on this kind of task without needing access to gradient information, as its memory store can provide the requisite information to update its predictions to the new task.

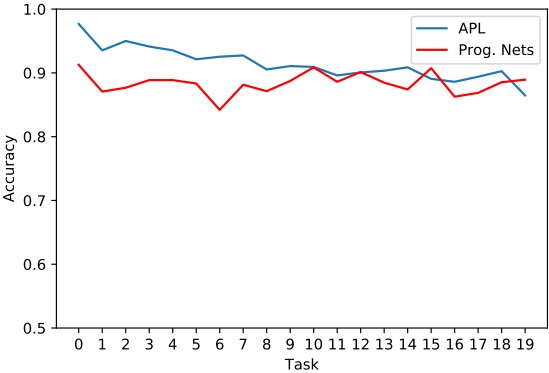

Figure 12: Accuracy of APL on a lifelong learning task where each task corresponds to learning 10 new classes in Omniglot. The baseline is a progressive net where the convolutional encoder is pretrained, and for every task a new logits layer is added and trained to classify the new classes. Results are the average accuracy over 5 runs. While not using any gradient information, APL performs as well or better than progressive networks.

