# OpenReview forum: "Adaptive Posterior Learning: few-shot learning with a surprise-based memory module"
_ICLR.cc/2019/Conference_

### Official Review · AnonReviewer1 · 2018-10-31
**Interesting algorithm for a few-shot learning**

**Rating:** 7
**Confidence:** 4

**Review:**

Summary: the authors propose a new algorithm, APL, for a few-shot and a life-long learning based on an external memory module. APL uses a surprise-based signal to determine which data points to store in memory and an attention mechanism to the most relevant points for prediction. The authors evaluate APL on a few-shot classification task on Omniglot dataset and on a number analogy task.

Quality: the authors consider interesting approach to life-long learning and I really liked the idea of a surprise-based signal to choose the data to store. However, I am not convinced by the learning setting that authors study. While a digit-symbol task from the introduction is interesting to study the properties of APL, I fail to see any real world analogy where it is useful. The same happens in a few-shot omniglot classification. The authors decided to shuffle the labels within episodes that, I guess, is supposed to represent different tasks in a typical life-long learning scenario. Again, it maybe interesting to study the behaviour of the algorithm, but I don't see any practical relevance here. It would make more sense to study the algorithm in a life-long learning setting, for example, considered in [1] and [2].

Clarity: the paper is well-written in general. I failed to decode the meaning behind the paragraph under Figure 3 on page 4 and would advise the authors to re-write it. The same goes to the first paragraph on page 3.

Originality: the paper builds on the prior work of Kaiser et al., 2017 and Santoro et al., 2016, but the proposed modifications are novel to my best knowledge.

Significance: below average: the paper combines interesting ideas that potentially can be used in different learning contexts and with other algorithms, however, the evaluation does not show the benefit in an obvious way.

Other comments:
* throughout the whole paper it is not clear if the embeddings are learned or not. I suppose they are, but what then happens to the ones in memory? If they are not, like in ImageNet example, where do they come from?
* the hyperparameter \sigma: the authors claim "the value of \sigma seems to not matter too much". Matter for what? It's great if the performance is stable for a wide range of \sigma, but it seems like it should have a great influence over the memory footprint of APL. I feel this is an important point that needs more attention.
* it would be interesting to see how APL performs with a simple majority vote instead the decoder layer. This would count for an ablation study and could emphasize the role of the decoder.
* Figure 4, b) plots are completely unreadable on black-and-white print, the authors might like to address that
* In conclusion, the first claim about state-of-the-art accuracy with smaller memory footprint: I don't think that the results of the paper justify this claim.

[1] Yoon et al, Lifelong Learning with Dynamically Expandable Networks, ICLR 2017
[2] Rebuffi et al,  iCaRL: Incremental Classifier and Representation Learning, CVPR 2017

********************
After authors response:

Thanks to the authors for a detailed response. The introduction led me to believe that the paper solves a different task from what it actually does. I still like the algorithm and, given that the scope of the paper is limited to a few-shot learning, I tend to change my evaluation and recommend to accept the paper. It was a good idea to change the title to avoid possible confusion by other readers. The introduction is still misleading though. It creates the impression that APL solves a more general problem where it would be good enough to limit the discussion to a few-shot learning setting and explain it in greater detail for an unfamiliar reader. Some details also seem to be missing, e.g. I didn't get that the memory is flushed after each episode and could not find where this is mentioned in the paper.

---

> ### Author Response · Authors · 2018-11-16
> **Response to reviewer 1**
>
> Thank you for the useful and constructive criticism which has helped us improve the paper. We address specific concerns below.
>
> >> “The authors decided to shuffle the labels within episodes that, I guess, is supposed to represent different tasks in a typical life-long learning scenario. Again, it maybe interesting to study the behaviour of the algorithm, but I don't see any practical relevance here.”
> We would like to stress that the tasks presented in the paper are not novel and arbitrary, but rather have been the subject of an extensive body of work in the meta-learning field [c.f. 1, 2, 3, 4, 5, 6, linked below, and further references on the related work section of this paper]. The motivation behind the label-shuffling task is a scenario where the model must learn to quickly associate high-dimensional data with a particular label (e.g. an image with a class label). Note that the models are always tested on a held-out test set of classes they have never seen before, which means in a real life scenario the model would be seeing a new class for the first time and would then immediately learn to associate subsequent data of the same class with the correct class.
>
> We emphasize that while few-shot learning is related to life-long learning, these are different research areas with different goals: few-shot learning focuses on doing well on a single task, where the model must perform well on new data not seen during training; while life-long learning focuses on adapting to *new* tasks not seen during training.
>
> >> “It would make more sense to study the algorithm in a life-long learning setting, for example, considered in [1] and [2].”
>
> While APL was devised to perform well in the few-shot learning scenario as explained above, we thank the reviewer for suggesting another interesting research area where APL could also be applied. We performed follow-up experiments to replicate the setup described in reference [1] provided by the reviewer and compared APL to progressive networks, a well-known life-long learning algorithm. We show that APL performs as well or better than progressive networks even though it does not need to perform gradient descent steps at test time. A thorough investigation of APL in the life-long learning setting would be out of scope for this paper but very interesting as follow up work!
>
> >> “I failed to decode the meaning behind the paragraph under Figure 3 on page 4 and would advise the authors to re-write it. The same goes to the first paragraph on page 3.”
>
> Thank you for these suggestions. We have rewritten these sections in the text in order to clarify their presentation.
>
> >> throughout the whole paper it is not clear if the embeddings are learned or not. I suppose they are, but what then happens to the ones in memory? If they are not, like in ImageNet example, where do they come from?
>
> The embeddings are learned in the case of omniglot and the digit analogy task. Each episode is short (we start with ~40*number of classes examples and anneal the episode length as accuracy improves), and the memory is flushed between each episode, so the embeddings in the memory are never too ‘old’.
>
> [1] Vinyals, Oriol, Charles Blundell, Timothy Lillicrap, Koray Kavukcuoglu, and Daan Wierstra. “Matching Networks for One Shot Learning.” ArXiv:1606.04080 [Cs, Stat], June 13, 2016. http://arxiv.org/abs/1606.04080.
> [2] Ren, Mengye, Eleni Triantafillou, Sachin Ravi, Jake Snell, Kevin Swersky, Joshua B. Tenenbaum, Hugo Larochelle, and Richard S. Zemel. “Meta-Learning for Semi-Supervised Few-Shot Classification.” ArXiv:1803.00676 [Cs, Stat], March 1, 2018. http://arxiv.org/abs/1803.00676.
> [3] Snell, Jake, Kevin Swersky, and Richard S. Zemel. “Prototypical Networks for Few-Shot Learning.” ArXiv:1703.05175 [Cs, Stat], March 15, 2017. http://arxiv.org/abs/1703.05175.
> [4] Finn, Chelsea, Kelvin Xu, and Sergey Levine. “Probabilistic Model-Agnostic Meta-Learning.” ArXiv:1806.02817 [Cs, Stat], June 7, 2018. http://arxiv.org/abs/1806.02817.
> [5] Nichol, Alex, Joshua Achiam, and John Schulman. “On First-Order Meta-Learning Algorithms.” ArXiv:1803.02999 [Cs], March 8, 2018. http://arxiv.org/abs/1803.02999.
> [6] Mishra, Nikhil, Mostafa Rohaninejad, Xi Chen, and Pieter Abbeel. “A Simple Neural Attentive Meta-Learner,” July 11, 2017. https://arxiv.org/abs/1707.03141.

---

> ### Author Response · Authors · 2018-11-16
> **Response to reviewer 1 (cont.)**
>
> >> the hyperparameter \sigma: the authors claim "the value of \sigma seems to not matter too much". Matter for what? It's great if the performance is stable for a wide range of \sigma, but it seems like it should have a great influence over the memory footprint of APL. I feel this is an important point that needs more attention.
>
> Thanks for the excellent question! We have added a detailed analysis of the behavior of the model as a function of \sigma in the supplementary information. To give you a quick summary \sigma does not affect the memory footprint of APL as much as might be assumed a priori as the model exhibits 2 regimes: before being completely trained, the model basically writes everything to memory; while after being trained the model is either completely surprised with a new data point (has never seen it before, or is significantly different from what it’s seen before), or classifies it with high accuracy. Therefore the model ends up writing roughly the same number of points to memory for a wide range of \sigma (obviously if you make it too high or too low the model breaks).
>
> >> it would be interesting to see how APL performs with a simple majority vote instead the decoder layer. This would count for an ablation study and could emphasize the role of the decoder.
>
> Matching networks can be seen as a special case of APL without a decoder and where all points are written to memory. Therefore the performance numbers for Matching Networks serve as an upper bound to the performance of APL ablated without a decoder. We will clarify this point in the text.
>
> >> Figure 4, b) plots are completely unreadable on black-and-white print, the authors might like to address that
>
> Thank you, we will strive to optimize the visual presentation of these plots in the final version of the paper.

---

### Official Review · AnonReviewer2 · 2018-10-31
**New idea in using memory for generalization task, more clarification and experiments are needed.**

**Rating:** 7
**Confidence:** 3

**Review:**

In this paper, authors present an algorithm to generalize learned properties from few observations by using a memory store and a memory controller. The experiments show comparable results on few-shot classification task and better performance and scalability for when the number of labels is unknown .

- The paper is well-written and easy to follow in general. The notations and model specifications are clear.

- The idea of incorporating an external memory store to save previous experiences is interesting especially without the need to backpropagate through the memory at each step. It is done by alignment of a query with the embeddings that are stored in the memory using k-nearest neighbor with Euclidean distance measures.  However, I am not quiet sure about how this is done in practice. It is stated in the paper that this alignment needs to emerge as a byproduct of training which is achieved by getting optimized to be as class-discriminative as possible. Isn't this implicitly optimizing part of the memory? I think more clarification would help a lot in understanding of this part.

-  I liked using a memory controller that decides whether a point is 'surprising'. Authors defined surprise to be negative log of prediction for label. I was wondering if they considered other measures, and investigated the effects that they might have. I think a brief discussion would be helpful.

- I am not an expert in this area but the experiments look convincing in general. Results in table one corresponding to 423-way are convincing since the proposed algorithm is the only candid that is able to perform the task with relatively good performance. On imagenet data set, the results are comparable to Inception-ResNet-v2 for fixed label case. However, more in-depth experiments or settings such as top-5 accuracy are needed to justify the performance of algorithm on this data set.  For the number analogy task the algorithm performs well in achieving high accuracy.

- Title of the paper is too generic. From the looks of it, adaptive posterior learning should cover wider set of tasks or probabilistic models, but it does not. So to avoid confusion (and the expectation that comes with this name), I strongly suggest that the authors change the title or make it more specific to actually represent what is discussed in the paper.

- In figure 4 c, I think x label should be "class number" not "number of classes".

---

> ### Author Response · Authors · 2018-11-16
> **Response to reviewer 2**
>
> Thanks a lot for your feedback. In the following we address some of the points you raised:
>
> - Representation alignment: Thank you for pointing this out. We have rewritten the corresponding section in the paper as this explanation could be made clearer. To quickly address your question, no gradients are calculated through the memory items. The weights of the encoder + decoder are optimized to minimize the cross-entropy loss for the current mini-batch alone, and then the embeddings produced by the encoder are stored in the memory (if the loss is high enough). Due to the nature of the classification problem, we expect embeddings for similar classes to be similar (in the euclidean distance case). Therefore the next time we see another example of that same class, the memory query should produce neighbors which share the same class. In this case, even though we never learn what to query or backpropagate through the memory, the query system should return the ‘correct’ set of neighbors. However this explanation is an intuitive hypothesis only and is not mathematically necessary! It could be the case that the encoder learns to produce very different embeddings for the same class, and therefore a k-nearest-neighbors query with euclidean distance would not return memories which are information. Which is why we needed to empirically verify whether the embeddings converge in the expected way or not. Our experimental results show that indeed this hypothesis is correct, and the query system works as we expected.
>
> - Alternative measure of surprise: We have added a discussion on this point as a general comment above.
>
> - Paper title: We agree that the title is quite broad and might lead to confusion amongst researchers from different areas. As a result we have extended it to better reflect the contents of the paper.
>
> Thanks again for the useful feedback!

---

### Official Review · AnonReviewer3 · 2018-11-02
**Review for adaptive posterior learning**

**Rating:** 6
**Confidence:** 4

**Review:**

The paper proposes a novel model that reads in information, decide whether this information is surprising and hence whether or not to keep it in memory and also utilizing information in the memory to quickly adapt or reason. The authors experimented with few-shot Omniglot classification and meta learning reasoning tasks.

Novelty:

The authors introduced a novel self-contained model that decides what to write to the external memory and making use of the external memory for different tasks.

My comments are mostly as follows:

1. The paper is well written, the problems are clearly stated, the solution is presented in a clear way, overall very easy to follow.

2. This is an interesting paper that combines a novel technique for writing to external memory based on surprisal and using it for more difficult tasks such as deductive reasoning.  I really like the surprisal mechanism, there are cognitive/ neuroscience materials that supports this approach (that the brain tends to write to memory things that are surprising). This also makes total sense from a machine learning perspective.

3. Could another objective  be used for surprisal? Also, instead of a determinstic encoder, decoder, is it possible to use a variational objective?

4. The experiments look convincing.

Overall a very nice paper, nice idea, could show more resul

---

> ### Author Response · Authors · 2018-11-16
> **Response to reviewer3**
>
> Thanks a lot for your comments and suggestions. In the following we address three of the points you raised:
>
> 1. Alternative measure of surprise: We have added a discussion on this point as a general comment above.
>
> 3. Variational objective. In this paper the main idea was to test the effectiveness of the memory controller mechanism coupled with a relational decoder. It is definitely possible to adapt a variational objective in the architecture and it would be a very interesting avenue for future work. Thank you for the suggestion.
>
> Additional experiments: as you suggested we have carried out more experiments to further consolidate our presentation of the model. We have applied APL to a set of continual learning experiments suggested by reviewer 3 and show that APL performs en par with progressive networks. These results are included the final version of the paper along with some pointers to the relevant literature.
>
> In light of the positive nature of your reviews we hope that these comments and the additional experiments can sway you to increase your rating of the paper.
> Thanks again for the useful comments!

---

### Author Response · Authors · 2018-11-16
**About surprise measure**

Several comments asked about alternatives to the surprise measure. We add a brief discussion below:

The proposed surprise measure uses the cross-entropy between the label prediction and the true label. This is equivalent to measuring how many more bits of information a perfect classifier would carry about the label as compared to the current model. In other words, this is the information we are missing about the label. To compress information ideally we want to store data which contains a lot of missing information, and discard redundant information. The approach we proposed simply thresholds on a provided number of bits derived from first principles. With more computational capacity it might be possible to optimize this threshold value via grid search, but the current value seems to provide good results.
Another option would be to use the classification accuracy as a proxy for surprise: if we make a mistake, then we store the data point. However this may not lead to optimal compression: suppose the case where two labels are very hard to distinguish and the correct posterior probabilities given an example from this vicinity would be [A:0.48, B:0.48, C:0.02, D:0.01, ...]. If the model predicts B instead of A in this case and we already have a few examples of A and B in memory, storing an additional example won’t help much as we will continue to make mistakes 50% of the time regardless of whether we store it or not. On the other hand, a measure based on the number of bits will be able to distinguish this case from one in which the model mistakenly places most of the probability mass on the wrong class, or simply outputs a uniform probability distribution.

It is also possible to learn a metric for surprise, for example by training a separate model which can tell us whether this input is surprising or not. This might be particularly useful in the case of unsupervised learning, where we don’t know how similar or dissimilar each data point is to the rest of the dataset. However, it is unclear whether this would help in the case of supervised learning, where there is already a natural low-dimensional representation of how examples relate to one another (i.e. the classes). Such an exploration would be interesting as future work.

---

### Public Comment · ~anony_anony_anony1 · 2018-11-30
**Can Memory store be updated?**

First, it is a very interesting idea.
I wonder if the Memory store can be updated? If a point stored in the Memory store, it will be deleted in the later iteration or stored in the memory forever.
Besides, is the Memory store with/without the upper limit?

Thanks.

---

### Public Comment · (anonymous) · 2018-12-22
**Nice take on adaptive learning**

This is a novel adaptive take on few-shot learning. It is great to see that it scales better and allows better generalization for < one-shot learning. I look forward to the released code.
I have three comments:
a) This work is a novel way to combine and generalize the techniques in [1] and [2], which I think are relevant works to discuss in the paper.
While out of scope currently, it would certainly be interesting
b) to compare different notions of surprise
c) assess the impact of stronger decoders such as in [3]

1 Shankar et al. Labeled Memory Networks for Online Model Adaptation. AAAI 2018
2 Sung et al. Learning to Compare: Relation Network for Few-Shot Learning. CVPR 2018
3 Bertinetto et al. Learning feed-forward one-shot learners. NIPS 2016

---

### Meta-Review · Area_Chair1 · 2018-12-13
**Strong paper**

**Confidence:** 5
**Recommendation:** Accept (Poster)

**Metareview:**

All reviewers recommend acceptance. The problem is an interesting one. THe method is interesting.
Authors were responsive in the reviewing process.

Good work. I recommend acceptance :)